# A randomized controlled trial of heterologous ChAdOx1 nCoV-19 and recombinant subunit vaccine MVC-COV1901 against COVID-19

Chih-Jung Chen[1,2,3], Lan-Yan Yang [ORCID][4], Wei-Yang Chang [ORCID][4], Yhu-Chering Huang[1,2,3], Cheng-Hsun Chiu[1,2,3], Shin-Ru Shih[5,6,7,8], Chung-Guei Huang[6,7] ✉ & Kuan-Ying A. Huang [ORCID][1,2,5,9] ✉

Heterologous prime-boost COVID-19 vaccine strategy may facilitate mass COVID-19 immunization. We reported early immunogenicity and safety outcomes of heterologous immunization with a viral vector vaccine (ChAdOx1) and a spike-2P subunit vaccine (MVC-COV1901) in a participant-blinded, randomized, non-inferiority trial (NCT05054621). A total of 100 healthy adults aged 20–70 years having the first dose of ChAdOx1 were 1:1 randomly assigned to receive a booster dose either with ChAdOx1 ($n = 50$) or MVC-COV1901 ($n = 50$) at an interval of 4–6 or 8–10 weeks. At day 28 post-boosting, the neutralizing antibody geometric mean titer against wild-type SARS-CoV-2 in MVC-COV1901 recipients (236 IU/mL) was superior to that in ChAdOx1 recipients (115 IU/mL), with a GMT ratio of 2.1 (95% CI, 1.4 to 2.9). Superiority in the neutralizing antibody titer against Delta variant was also found for heterologous MVC-COV1901 immunization with a GMT ratio of 2.6 (95% CI, 1.8 to 3.8). Both spike-specific antibody-secreting B and T cell responses were substantially enhanced by the heterologous schedule. Heterologous boosting was particularly prominent at a short prime-boost interval. No serious adverse events occurred across all groups. The findings support the use of heterologous prime-boost with ChAdOx1 and protein-based subunit vaccines.

The Oxford-AstraZeneca COVID-19 vaccine, also known as AZD1222 or ChAdOx1 nCoV-19 (ChAdOx1), was one of the earliest authorized and the most widely used vaccines (in 182 countries until January 2022) during the global fighting against COVID-19[1–3]. It has been demonstrated in the clinical trials that ChAdOx1 given at a two-dose schedule was of 70.4% efficacy against laboratory-confirmed symptomatic COVID-19[4,5]. Although with lower efficacy compared to the mRNA and protein-based vaccines, the real-world effectiveness data further revealed that ChAdOx1 with the original regimen was highly effective against severe COVID-19 diseases and fatal outcomes

caused by the dominant variants of concern of SARS-CoV-2 virus[6,7]. Unfortunately, within three months of deployment, a public concern of the safety of ChAdOx1 was abruptly raised due to its linkage to a rare but potentially lethal blood clot disorder termed thrombosis and thrombocytopenia syndrome (TTS) or vaccine-induced thrombotic thrombocytopenia[8–10]. Replacement of the second dose with non-adenovirus vector vaccine was considered as an alternative immunization strategy against COVID-19 in ChAdOx1 recipients at risk of TTS. The consideration was later supported by clinical trials and observational studies on mix-and-match strategy displaying

comparable safety profiles, enhanced and broadened immunogenicity, and improved clinical effectiveness against COVID-19 among ChAdOx1 recipients boosted with mRNA vaccines including BNT162b2 (Comirnaty, BioNTech/Pfizer) and mRNA-1273 (Spikevax, Moderna)[11–15]. The heterologous schedule with vaccines from different platforms was widely adopted and officially recommended in many countries as the acceptable immunization strategy against COVID-19.

The usefulness of the protein-based COVID-19 vaccine as the second shot to the adenovirus-vector vaccine recipients has been previously evaluated in a few clinical trials to our knowledge[16]. COVID-19 vaccines from the two manufacturing platforms share similar characteristics including good thermostability, easy storage, transportation and are suitable for deployment, especially in resource-limited regions. We set up a pilot study to evaluate the reactogenicity and immunogenicity of the heterologous prime-boost vaccination strategy with the ChAdOx1 as the first dose and the MVC-COV1901 as the booster dose. The results including the antibody responses to the ancestral Wuhan strain and the predominant strain (Delta variant) in 2021 after the booster dose were presented in this report.

MVC-COV1901 was a CpG 1018- and aluminum hydroxide-adjuvanted recombinant subunit vaccine containing pre-fusion-stabilized spike protein S-2P developed by Medigen. It has been demonstrated the advantage of S-2P conformation in the designation of a vaccine against coronaviruses in both the immunogenicity and protective efficacy in animal models[17,18]. MVC-COV1901 was officially authorized for emergency use in July 2021 in Taiwan after a large-scale phase 2 trial with more than four-thousand participants demonstrating a good safety profile and promising immunogenicity[16]. It was the first S-2P protein-based vaccine being deployed against COVID-19 in the world. The clinical efficacy of MVC-COV1901 is under-evaluated in a global, randomized, placebo-controlled, phase 3 trial by World Health Organization (the Solidarity Trial Vaccines) since late 2021[19].

## Results

### The participants

A total of 100 one-dose ChAdOx1 participants at the ages of 22 to 62 years (median and mean ages, 40 and 40.9 years, respectively) were 1:1 randomly assigned to receive ChAdOx1 ($n = 50$) or MVC-COV1901 ($n = 50$) as the booster dose. The demographics, baseline vital signs, baseline laboratory values, and intervals between prime and boost vaccines were well balanced between both groups (Supplementary Table 1). Most of the participants were healthy Han Taiwanese without major systemic disorders. Type 2 diabetes and thyroid function disorders under medical control were respectively reported by three subjects (Supplementary Table 1). The other minor underlying conditions are listed in Supplementary Table 2. No withdrawals have occurred before the analysis at day 28 of booster vaccination (Fig. 1).

### Safety and tolerability

The common solicited AEs occurring within one week after the boost dose for all recipients were pain at the injection site (63.0%), fatigue (43.0%), headache (28.0%), and myalgia (27.0%). The incidences of the common solicited AEs did not differ significantly between the two groups (Fig. 2 and Supplementary Table 3). However, the AEs of high-grade severity tended to be more commonly identified in ChAdOx1 recipients. Of them, the greater incidence of high-grade fatigue (≥ grade 2 severity) in the ChAdOx1 recipients than in MVC-COV1901 reached statistical significance (18.0% versus 6.0%, $P = 0.0160$, Supplementary Table 3 and Fig. 2). There was no case with serious AE in both groups before day 28 of booster vaccination in this analysis.

### Neutralizing antibody response

Neutralizing antibody (nAb) titer was assayed using a surrogate ELISA-based assay[18]. Before administration of the booster dose, the baseline

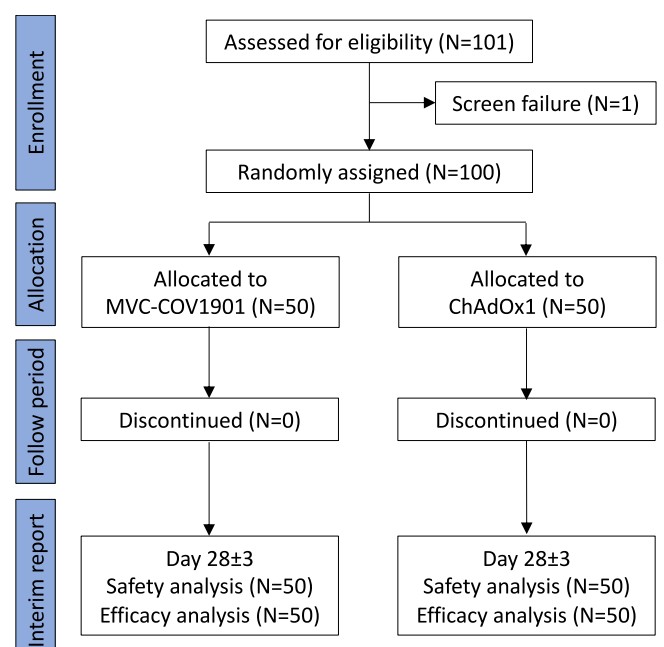

**Fig. 1 | Consort diagram of study design.** A diagram showing trial groups and participant flow in the study (NCT05054621). Participants were randomly assigned to receive either a heterologous boost of MVC-COV1901 or a homologous booster dose of ChAdOx1. There was one screen failure (participant S30017 was unable to visit the study site in the scheduled time points) in the trial.

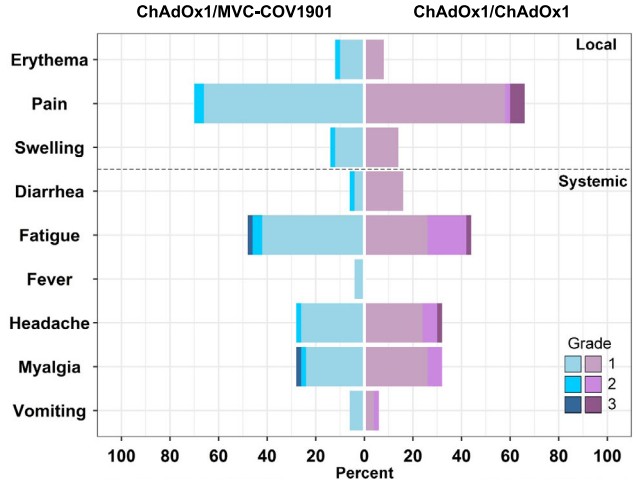

**Fig. 2 | Solicited local and systemic adverse events in the 7 days after the booster dose of the heterologous ChAdOx1/MVC-COV1901 ($n = 50$) and homologous ChAdOx1/ChAdOx1 ($n = 50$) group.** Grades 1, 2 and 3 adverse events were marked in light blue, blue and deep blue for the ChAdOx1/MVC-COV1901 group. Grades 1, 2 and 3 adverse events were marked in light purple, purple and deep purple for the ChAdOx1/ChAdOx1 group. The higher the grade, the more severe the adverse event. Source data are provided as a Source Data file.

nAb titers were at low levels, with similar GMT of 32.2 IU/mL and 30.2 IU/mL respectively for day 28 ± 3 vaccine recipients of two groups ($P = 0.7900$, Fig. 3a and Supplementary Table 4). After the booster dose, the nAb GMT significantly elevated to 202.1 IU/mL at day 10 ± 3 (95% confidence interval [CI], 162.1–252.1 IU/m) and 235.5 IU/mL at day 28 ± 3 (95% CI, 186.7–297.1 IU/mL) in recipients boosted with MVC-COV1901, which were 2.6-fold (95% CI, 1.7–4.0 folds) and 2.1-fold (95% CI, 1.4–2.9 folds) higher than in those boosted with ChAdOx1 at the two respective time points (two-tailed Mann–Whitney, $P < 0.0010$ for day 10 ± 3; $P < 0.0010$ for day 28 ± 3) (Fig. 3a and Supplementary Table 4).

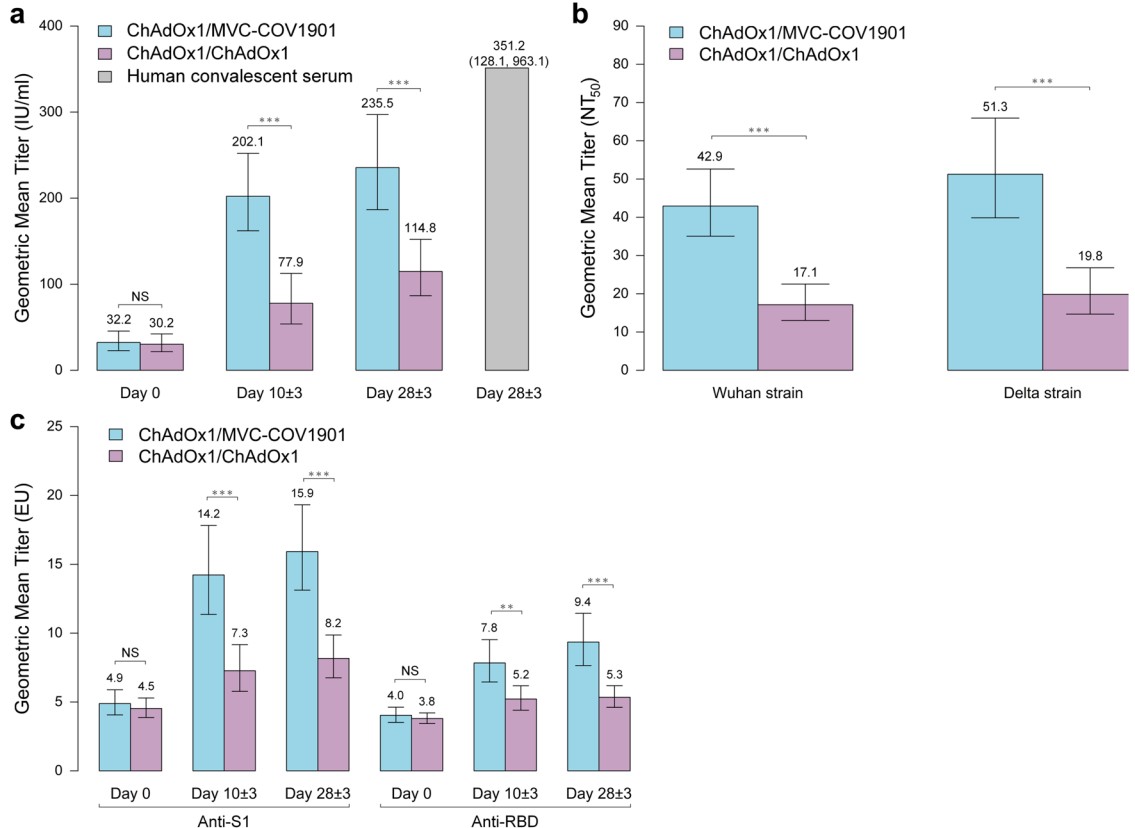

**Fig. 3 | Comparison of antibody titers between heterologous ChAdOx1/MVC-COV1901 ($n = 50$) and homologous ChAdOx1/ChAdOx1 ($n = 50$) groups prior to and after booster vaccination. a** Neutralizing antibody titers measured by ELISA-based method. The data from 15 human convalescent serum samples collected at day $28 \pm 3$ after diagnosis is shown for comparison (grey bar). The values in international units per milliliter (IU/mL) are provided. Data are presented as geometric mean ±95% confidence intervals. The significance between two groups was determined using $t$ test (two-tailed). Comparisons of antibody titers, Day 0, $P = 0.7900$; Day $10 \pm 3$, $P < 0.0010$; Day $28 \pm 3$, $P < 0.0010$. **b** Neutralizing antibody titers against wild-type Wuhan-1 and Delta variant measured by plaque reduction neutralization assay for day $28 \pm 3$ serum. The half-maximal neutralization titer ($NT_{50}$) values are provided. Data are presented as geometric mean ±95% confidence intervals. The significance between two groups was determined using $t$ test (two-tailed). Comparisons of antibody titers, Wuhan strain, $P < 0.0010$; Delta strain, $P < 0.0010$. **c** Binding-antibody titers in ELISA Unit (EU) against spike S1 protein and receptor-binding domain (RBD). Comparisons of anti-S1 titers, Day 0, $P = 0.5260$; Day $10 \pm 3$, $P < 0.0010$; Day $28 \pm 3$, $P < 0.0010$. Comparisons of anti-RBD titers, Day 0, $P = 0.4850$; Day $10 \pm 3$, $P = 0.0020$; Day $28 \pm 3$, $P < 0.0010$. Data are presented as geometric mean ±95% confidence intervals. The significance between two groups was determined using $t$ test (two-tailed). **$P < 0.01$, ***$P < 0.001$, NS not significant. Source data are provided as a Source Data file.

Neutralizing titer of day $28 \pm 3$ serum against live SARS-CoV-2 virus of wild-type Wuhan and Delta variant were measured with plaque reduction neutralization assay. Consistent with the finding by ELISA method, the live virus nAb titers against wild-type Wuhan and Delta variant were both higher in recipients boosted with MVC-COV1901 than those boosted with ChAdOx1, with GMT ratios of 2.5-fold (95% CI, 1.8–3.5 folds) and 2.6-fold (95% CI, 1.8–3.8 folds), respectively (two-tailed Mann–Whitney, $P < 0.001$ for wild-type Wuhan; $P < 0.001$ for Delta variant) (Fig. 3b and Supplementary Table 4). When comparing the results of nAb titers generated by live virus neutralization and the ELISA, we found a high degree of correlation between the assays, with $R^2$ values of 0.718 against the wild-type Wuhan and 0.663 against Delta variant (Supplementary Table 5).

**Spike S1- and RBD-binding antibody responses**
The binding antibody (bAb) titers against S1 protein and RBD in both groups are shown in Fig. 3c and Supplementary Table 4. Consistent with the trend observed for nAb, the RBD-binding antibody titer was higher in recipients boosted with MVC-COV1901 than those boosted with ChAdOx1, with GMT ratios of 1.5-fold and 1.8-fold at day $10 \pm 3$ and day $28 \pm 3$, respectively (two-tailed Mann–Whitney, $P = 0.0020$ for day $10 \pm 3$; $P < 0.0010$ for day $28 \pm 3$). Similar result is observed for

the S1-binding antibody response (Fig. 3c and Supplementary Table 4).

**Association of dose intervals and nAb titers**
There was no significant difference in the baseline nAb titer between heterologous and homologous groups for each prime-boost interval stratum (Supplementary Table 6). For the heterologous MVC-COV1901 group, there was no significant difference in the nAb titer at baseline between two prime-boost interval strata (Supplementary Table 6). After boosting, the recipients with short prime-boost interval (4–6 weeks) had higher nAb titers compared to those with long interval (8–10 weeks) at day $10 \pm 3$ (GMT, 258.4 IU/mL versus 158.2 IU/mL, $P = 0.0250$) and day $28 \pm 3$ (GMT, 325.3 IU/mL versus 170.5 IU/mL, $P = 0.0050$) (Fig. 4 and Supplementary Table 6) and those with short interval had higher nAb titer change from baseline at day $28 \pm 3$ (Supplementary Fig. 1).

A similar trend of higher nAb titers favoring the short interval was also identified in the homologous ChAdOx1 group at day $10 \pm 3$ though the difference of GMTs did not reach statistical significance at day $28 \pm 3$ (134.4 IU/mL versus 98.0 IU/mL, $P = 0.2630$, Fig. 4 and Supplementary Table 6). No significant difference in nAb titer change from baseline was observed between subgroups with short and long intervals at day $10 \pm 3$ and day $28 \pm 3$ (Supplementary Fig. 1).

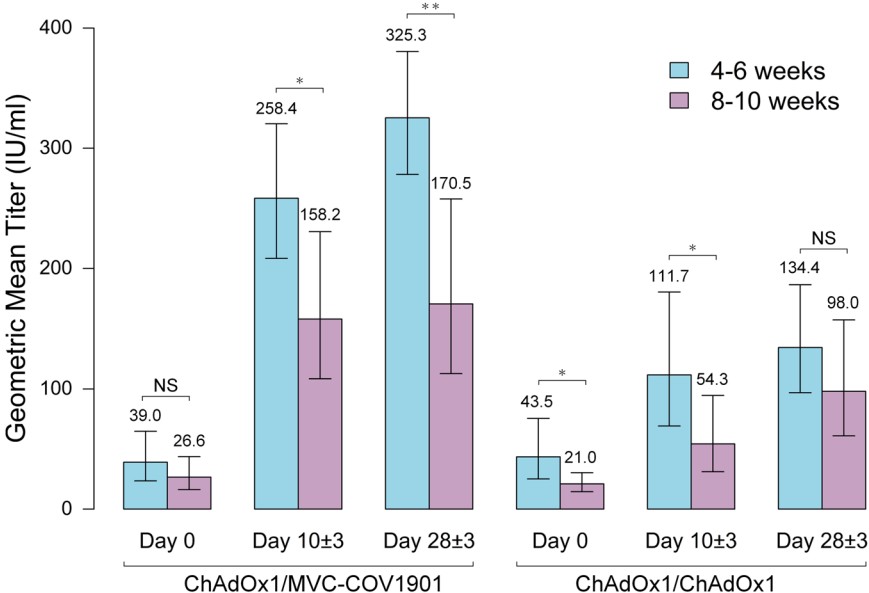

**Fig. 4 | Comparisons of antibody titers of short (4–6 weeks, *n* = 25) and long (8–10 weeks, *n* = 25) vaccine dosing intervals.** Neutralizing antibody titers were measured by ELISA-based method. The values in international units per milliliter (IU/mL) are provided. Data are presented as geometric mean ±95% confidence intervals. The significance between two dose intervals was determined using *t* test (two-tailed). Comparisons of antibody titers for the heterologous MVC-COV1901 group, Day 0, *P* = 0.2720; Day 10 ± 3, *P* = 0.0250; Day 28 ± 3, *P* = 0.0050. Comparisons of antibody titers for the homologous ChAdOx1 group, Day 0, *P* = 0.0270; Day 10 ± 3, *P* = 0.0480; Day 28 ± 3, *P* = 0.2630. *P < 0.05, **P < 0.01, NS not significant. Source data are provided as a Source Data file.

## Spike-specific antibody-secreting B cell response

We assessed the levels of SARS-CoV-2 spike-specific antibody-secreting B cells in the peripheral blood over the course of booster vaccination. The predominance of an IgG response following the booster dose was observed in both groups of MVC-COV1901 and ChAdOx1 recipients by assessment of spike binding of IgG-, IgM- and IgA-secreting B cells by ex vivo ELISPot (Fig. 5a, b).

Significant increase in the frequency of spike-specific IgG-secreting cells was observed on day 10 ± 3 after the booster dose in both groups (one-way ANOVA, *P* < 0.0001 for either group). Such IgG response was accompanied with a lower IgA-secreting cell response while IgM-secreting cell response was barely detectable in both groups (one-way ANOVA, *P* < 0.0001 for either group). The frequency of day 10 ± 3 spike-specific IgG-secreting cells was significantly higher in MVC-COV1901 recipients than that in ChAdOx1 recipients (two-tailed Mann–Whitney, *P* = 0.0007). Similar result is observed for IgA-secreting cell response (two-tailed Mann–Whitney, *P* = 0.0010), but no significant difference was observed for IgM response (Fig. 5b).

The frequency of spike-specific IgG-secreting cells was significantly correlated with the surrogate neutralizing titer measured by ELISA method on day 10 ± 3 (*P* = 0.0002) and 28 ± 3 (*P* = 0.0024) after the booster dose (Fig. 5c).

Day 10 ± 3 spike-specific antibody-secreting cell response was compared between short (4–6 weeks) and long (8–10 weeks) vaccination intervals. A strongest IgG-secreting cell response was detected in the subgroup of MVC-COV1901 recipients with short vaccination interval (one-way ANOVA, *P* = 0.0004) (Fig. 5d). No significant changes were seen in the frequency of IgM- and IgA-secreting cells among subgroups (Fig. 5d).

## Spike-specific T cell response

SARS-CoV-2-specific T cell responses to spike antigens were measured by ex vivo IFN-γ-ELISpot prior to, on day 10 ± 3, and day 28 ± 3 after their booster dose of ChAdOx1 or MVC-COV1901 vaccines (Fig. 6a). T cell responses were detected prior to and during the course of vaccination for all recipients (Fig. 6b, c). A boosting effect was observed in ChAdOx1/MVC-COV1901 recipients on days 10 ± 3 (one-way ANOVA,

*P* < 0.0001) and 28 ± 3 (one-way ANOVA, *P* = 0.0486). On day 28 ± 3, the spike-specific T cell response had contracted from the peak response but remained nearly 1.5-fold higher than that prior to the booster dose in ChAdOx1/MVC-COV1901 recipients. By contrast, no significant boosting effect was observed in ChAdOx1/ChAdOx1 recipients (Fig. 6b)

ChAdOx1/MVC-COV1901 recipients had significantly higher spike-specific T cell responses than ChAdOx1/ChAdOx1 recipients on days 10 ± 3 (two-tailed Mann–Whitney, *P* = 0.0039) and 28 ± 3 (two-tailed Mann–Whitney, *P* = 0.0053) (Fig. 6b). These stronger T cell responses were mapped to both S1 and S2 antigens for both day 10 ± 3 and day 28 ± 3 time points (Fig. 6b, c and Supplementary Fig. 2). The S1 subunit responses was higher than S2 subunit response for both groups of recipients, but the difference was not significant (Fig. 6c and Supplementary Fig. 2).

We compared the fold change of T cell responses after the booster dose between short (4–6 weeks) and long (8–10 weeks) vaccination regimens. Vaccine recipients that had their 1st dose of ChAdOx1 4–6 weeks before had a significantly higher T cell response than those had their 1st dose 8–10 weeks ago (one-way ANOVA, *P* = 0.0013) (Supplementary Fig. 3). On day 10 ± 3, no significant difference in fold change of T cell response was observed between subgroups with short and long vaccination intervals (Fig. 6d). On day 28 ± 3, a significantly higher fold change of T cell response was detected in the subgroup of ChAdOx1/MVC-COV1901 recipients with long vaccination interval than the subgroup of ChAdOx1/ChAdOx1 recipients with short vaccination interval (one-way ANOVA, *P* = 0.0234) (Fig. 6d).

## Correlation between T cell response and fold increase in spike-specific response and antibody response post vaccination

The impact of pre-existing spike-specific T cells on induction of T cell responses post vaccination was next investigated. An inverse correlation was found between spike-specific T cell responses at baseline and the fold increase in spot-forming cells in IFN-γ-ELISpot post-vaccination (Supplementary Fig. 4a). The inverse correlations between total spike-specific T cell response in the baseline and fold-change post-vaccination were strongly statistically significant after the

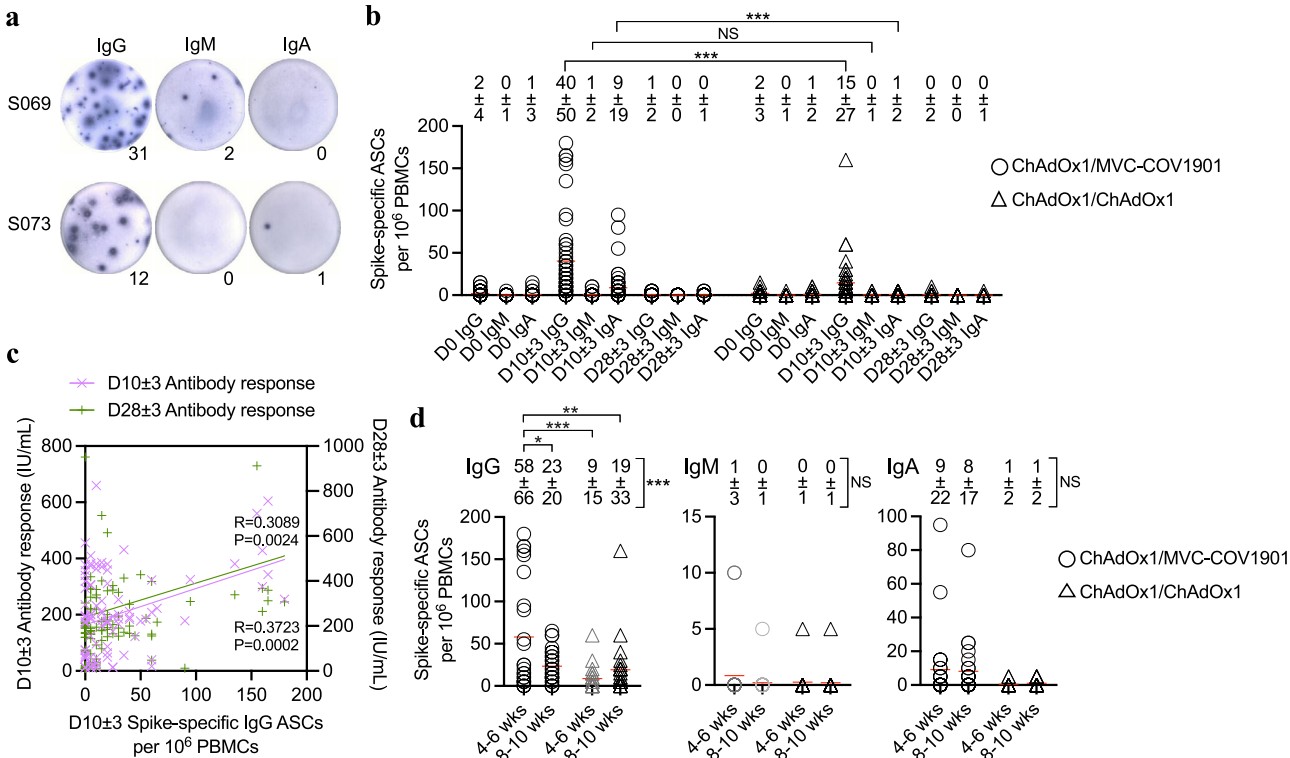

**Fig. 5 | Antibody-secreting B cell response to SARS-CoV-2 spike is detected in recipients after the booster dose. a** Representative ELISpot from one ChAdOx1/MVC-COV1901 recipient (S069) and one ChAdOx1/ChAdOx1 recipient (S073). **b** IgG, IgM, IgA-secreting ASCs frequencies to spike prior to, at day 10 ± 3, and day 28 ± 3 after the booster dose in ChAdOx1/MVC-COV1901 (*n* = 49) and ChAdOx1/ChAdOx1 (*n* = 45) groups. The mean frequency of spike-specific ASCs response and its standard deviation at each time point is shown in the figure. Each point represents a single recipient and red line represents the mean. The significance between two groups was determined using a Mann–Whitney test (two-tailed). Day 10 ± 3 IgG, *P* = 0.0007; Day 10 ± 3 IgM, *P* = 0.7674; Day 10 ± 3 IgA, *P* = 0.0010. **c** The relationship between day 10 ± 3 IgG-secreting ASCs response and serological antibody response at day 10 ± 3 (*n* = 94) and day 28 ± 3 (*n* = 94). Antibody titer was measured using a SARS-CoV-2 surrogate virus neutralization test and represented as international units (IU) per mL of serum. The correlation was determined using simple linear regression. **d** Comparison of day 10 ± 3 ASCs response in short (4-6 weeks, *n* = 24 in ChAdOx1/MVC-COV1901 group, *n* = 20 in ChAdOx1/ChAdOx1 group) and long (8–10 weeks, *n* = 25 in ChAdOx1/MVC-COV1901 group, *n* = 25 in ChAdOx1/ChAdOx1 group) vaccine dosing intervals. The mean frequency of spike-specific ASCs response and its standard deviation is shown in the figure. Each point represents a single recipient and red line represents the mean. The significance between vaccine dosing intervals was determined using one-way ANOVA with post hoc Dunn's multiple comparison test. IgG, *P* = 0.0004, post hoc ChAdOx1/MVC-COV1901 short vs ChAdOx1/MVC-COV1901 long, *P* = 0.0144; ChAdOx1/MVC-COV1901 short vs ChAdOx1/ChAdOx1 short, *P* = 0.0004; ChAdOx1/MVC-COV1901 short vs ChAdOx1/ChAdOx1 long, *P* = 0.0046. IgM, *P* = 0.4917. IgA, *P* = 0.0657. **P* < 0.05, ***P* < 0.01, ****P* < 0.001, NS not significant, ASC antibody-secreting B cell. Source data are provided as a Source Data file.

booster dose in all recipients (*P* = 0.0395 for day 10 ± 3 fold change, *P* = 0.0308 for day 28 ± 3 fold change) (Supplementary Fig. 4a).

Positive correlations were found between spike-specific T cell response on day 10 ± 3 and peak antibody-secreting B cell response (*P* < 0.0001) and serological antibody response (*P* = 0.0044) on day 28 ± 3 (Supplementary Fig. 4b). Similar correlation results were observed for day 28 ± 3 T cell response (Supplementary Fig. 4c).

## Discussion

Results from the study demonstrated that the heterologous prime-boost schedule with ChAdOx1 followed by MVC-COV1901 in healthy adult subjects elicited significantly greater humoral and cellular immunogenicity when compared to the homologous schedule with two doses of ChAdOx1 vaccination. The observed increase in immunogenicity for the heterologous schedule was particularly prominent when the prime and boost vaccines were administered at a short interval between 4 to 6 weeks. This study has shown that the protein-based subunit vaccine MVC-COV1901 was able to evoke a strong booster response in recipients primed with the adenovirus-vector vaccine and may provide better protection against the ancestral and Delta variant of SARS-CoV-2 virus than a ChAdOx1 vaccine boost.

The reactogenicity profiles were generally consistent with the safety data published for the homologous schedule of both vaccines in their respective clinical trials[4,5,16,23]. Although the incidences of most local and systemic AEs irrespective of severity did not differ with statistical significance between the two groups, the AEs of grade 3 severity including pain at the injection site and headache were exclusively identified in the participants on homologous schedule. The only headache event of grade 3 severity occurred in a ChAdOx1 recipient after the booster shot and persisted for three days which required analgesic treatment in the emergency department. The fatigue of grade 2 severity was also significantly more common after the ChAdOx1 vaccination. It has been reported that the booster shot was associated with lower incidences of AEs compared to the prime dose for the vaccine at a homologous schedule[4,5]. The observation on safety data in this head-to-head comparison study further demonstrated a preferable reactogenicity profile for the heterologous schedule with ChAdOx1 followed by MVC-COV1901.

A growing body of evidence has demonstrated the superiority of heterologous prime-boost regimes with ChAdOx1 followed by mRNA vaccines over the homologous two-dose ChAdOx1 vaccination in both the humoral and cellular immune responses against SARS-CoV-2. On day 28 after a boost, the anti-spike IgG titer and live virus nAb titer were 9.2-fold and 6.4-fold higher in heterologous ChAdOx1–BNT162b2 recipients than in two-dose ChAdOx1 recipients in a UK study[24]. The frequencies of spike-specific T cells were also significantly higher in the

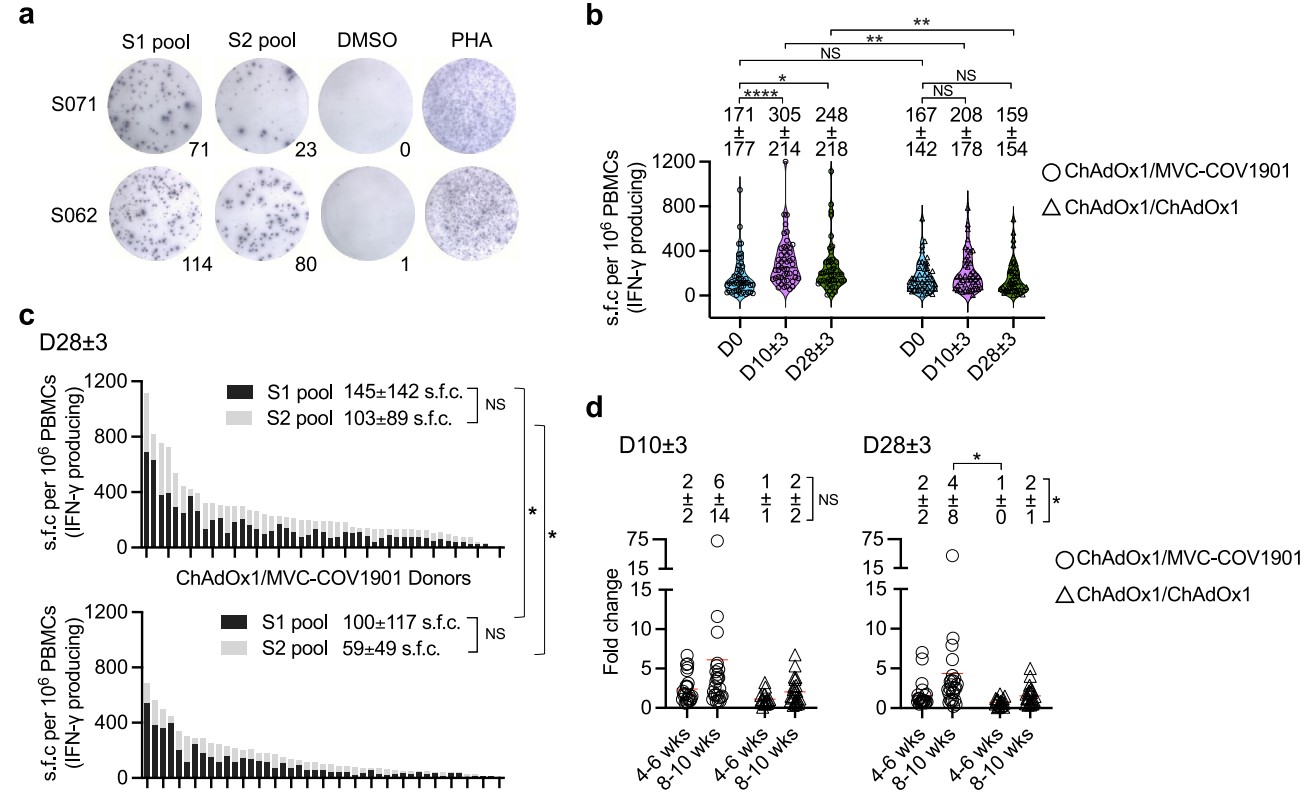

**Fig. 6 | T cell response to SARS-CoV-2 spike is present in recipients prior to and after the booster dose. a** Representative ELISpot from one ChAdOx1/MVC-COV1901 recipient (S062) and one ChAdOx1/ChAdOx1 recipient (S071) against spike (pools 1 and 2), with DMSO as negative control and PHA as positive control. **b** Total spike-specific T-cell responses (sum of S1 and S2 subunit responses, mean ± standard deviation) prior to, at day 10 ± 3, and day 28 ± 3 after the booster dose in ChAdOx1/MVC-COV1901 (n = 49) and ChAdOx1/ChAdOx1 (n = 45) groups. Each point on violin plot represents a single recipient and black line represents the median. The significance was determined using one-way ANOVA with post hoc Dunn's test. ChAdOx1/MVC-COV1901, Day 0 vs Day 10 ± 3, P < 0.0001; Day 0 vs Day 28 ± 3, P = 0.0486. ChAdOx1/ChAdOx1, Day 0 vs Day 10 ± 3, P > 0.9999; Day 0 vs Day 28 ± 3, P > 0.9999. The significance between two groups was determined using a Mann–Whitney test (two-tailed). Day 0, P = 0.7022; Day 10 ± 3, P = 0.0039; Day 28 ± 3, P = 0.0053. **c** Summary data of day 28 ± 3 T cell response in vaccine

recipients according to spike peptide pools (mean±standard deviation). The significance was determined using one-way ANOVA with post hoc Dunn's test. S1 pool comparison, P = 0.0345; S2 pool comparison, P = 0.0334. **d** The fold change of total spike-specific T-cell responses on day 10 ± 3 (left panel) and 28 ± 3 (right panel) relative to the response prior to the booster dose in short (4–6 weeks, n = 24 in ChAdOx1/MVC-COV1901 group, n = 20 in ChAdOx1/ChAdOx1 group) and long (8–10 weeks, n = 25 in ChAdOx1/MVC-COV1901 group, n = 25 in ChAdOx1/ChAdOx1 group) vaccine dosing intervals (mean±standard deviation). Each point represents a single recipient and red line represents the mean. The significance was determined using one-way ANOVA with post hoc Dunn's test. Day 28 ± 3, P = 0.0234, post hoc ChAdOx1/MVC-COV1901 long vs ChAdOx1/ChAdOx1 short, P = 0.0273. *P < 0.05, **P < 0.01, ****P < 0.0001, NS not significant, DMSO dimethylsulfoxide, PHA Phytohaemagglutinin, s.f.c. spot-forming cells. Source data are provided as a Source Data file.

heterologous group after the booster dose[24]. A similar observation on the stronger immune response evoked by a heterologous regimen with ChAdOx1 followed by the Moderna mRNA vaccine (mRNA-1273) was also reported in the healthcare workers in Sweden[14]. When compared to homologous ChAdOx1 vaccination, the titers were approximately 23-fold and 10-fold higher for anti-spike IgG and nAb, respectively, after the mRNA-1273 boost. However, the enhanced immunogenicity of heterologous regimen with mRNA vaccine boost was generally associated with greater incidences of AEs[14,24]. The use of a protein-based subunit vaccine (VX-CoV2373 by Novavax) as a booster to the ChAdOx1 was recently reported in a UK study (Com-COV2)[15]. The anti-spike IgG titer was significantly elevated after the VX-CoV2373 booster though with a relatively smaller magnitude (2.8 folds) compared to the mRNA-1273 booster (10.2 folds)[15]. In line with our finding on heterologous boost with MVC-COV1901, the ChAdOx1–VX-CoV2373 heterologous schedule was not associated with increased incidences in systemic or local AEs.

The ELISA method for measuring nAb was developed by our group based on the binding affinity of antibodies to both the S1 protein and the receptor-binding domain. Comparing the results of surrogate nAb of day 28 serum using the ELISA method with those of live virus

neutralization assay, we observed a high degree of correlation between both assays. A similar magnitude of fold increase in neutralizing activity in recipients on heterologous schedule versus those on the homologous schedule was also consistently demonstrated by the two assays. Taken together, the finding suggested that the binding antibody-based ELISA method was a robust assay in the prediction of neutralizing activity to SARS-CoV-2 and was useful especially in large scales studies given its automation characteristic.

The finding that short dose intervals of 4 to 6 weeks tended to be associated with higher nAb titers was unexpected especially for the group on the homologous schedule. In the initial phase 1/2 trial of the ChAdOx1 vaccine, both the nAb titers and bAb titers including IgG, IgG1 and IgG3 against SARS-CoV-2 were not significantly different between groups respectively boosted at intervals of 28 days and 56 days with the standard dose of ChAdOx1[25]. Another pooled analysis of four phase 2/3 trials of the ChAdOx1 vaccine displayed a greater vaccine efficacy in recipients with a long prime-boost interval of ≥12 weeks than in those with a short interval of ≤ 6 weeks (efficacy, 81.3% versus 55.1%)[26]. In line with the data of clinical efficacy, the pseudovirus neutralization level and bAb response at day 28 post booster dose were positively associated with an increased interval

from <6 weeks to ≥12 weeks[26]. The reason for the conflict results between our study and previous observations remained unclear. During the period of this heterogenous schedule trial, there was an extremely low incidence of indigenous COVID-19 cases in Taiwan. It was different in the nations including the UK, Brazil and South Africa where the epidemic waves were occurring when the phase 2/3 trials of the ChAdOx1 vaccine were conducted. The possibility of immunogenicity data confounded by natural infections cannot be completely excluded. Further, the effect of interval on immunogenicity was an exploratory outcome in the pooled analysis of the phase 2/3 trials. In the current study, the dosing interval was an important parameter that was well controlled and the favourable neutralizing activity toward short interval was confirmed in the experiments on spike-specific antibody-secreting B cell response. However, it was noteworthy that the significance of increased immunogenicity for the short interval was lost at day 28 post the booster dose in recipients on homologous schedule but remained in those on heterologous schedule. The ongoing follow-up immunogenicity data on day 56 and 168 post the booster dose in this trial will add further insight into the impact of interval and schedule on the kinetics of the vaccine-evoked immunogenicity.

We showed that both vaccines, which have proven to be highly immunogenic[4,16,23], induced spike-specific T cells against both S1 and S2 antigens after boosting. Spike-specific T cell responses to ChAdOx1 could be primarily mediated by Th1-dominated CD4 + T cell helper type 1 and also CD8 + T cells that might help control or prevent SARS-CoV-2[27]. A stronger T cell response to heterologous vaccination primed with ChAdOx1 has been reported[13,25]. This was associated with a broader antibody response that cross-reacts with other variants[28], which is in line with our findings of antibody profiles after heterologous boosting.

A rapid elicitation of IgG dominated antibody-secreting B cell response was detected after boosting in both heterologous and homologous prime-boost groups, indicative of the generation of secondary immune response. Nevertheless, stronger spike-specific antibody-secreting B cell response with higher levels of neutralizing and spike-binding antibodies were observed in the heterologous group. Similar results were found in the study of ChAdOx1/mRNA-1273 prime-boost vaccination[28]. Extended studies are needed to understand the breadth and function of antibody repertoire derived from spike-specific B cell population among heterologously immunized individuals. It would be of great importance to formulate optimal vaccination strategy to achieve protective immunity against emerging variants in the near future.

Although a relatively small number of participants were enrolled, the trial results of vaccine-elicited immunity demonstrate strong neutralizing antibody and spike-specific cellular responses after a heterologous ChAdOx1 and MVC-COV1901 immunization. Importantly, this heterologous booster immunization is well tolerated. Recent studies have shown the safety and non-inferior immunogenicity of combination of ChAdOx1 and mRNA vaccines[12–14], subunit vaccine[15] or inactivated vaccines[29]. The current data support the use of heterologous prime-boost vaccination with ChAdOx1 and MVC-COV1901 vaccines.

## Methods
### Study design
This was an investigator-initiated, single-blinded, 1:1 randomized vaccine clinical study, designed to assess the reactogenicity and immunogenicity of heterologous prime-boost immunization with ChAdOx1 (AZD1222) followed by the subunit vaccine MVC-COV1901, compared with the homologous immunization with two doses of ChAdOx1 (Fig. 1). Participants were healthy adults without severe disorders at the age of 20–70 years who have had their first dose of the ChAdOx1 vaccine. There was no TTS or other serious adverse events following the first ChAdOx1 vaccination in all participants. For female participants, they must be either of non-childbearing potential (i.e., surgically

sterilized or one year post-menopausal) or, if of childbearing potential, be abstinent or agree to use medically effective contraception on enrollment continuously until 90 days after boost immunization of study intervention. A negative pregnancy test was required before enrollment.

The primary objective of this trial was to determine if the immune response (neutralizing antibody titer at day 28 after the booster dose) of heterologous group was non-inferior to that observed in the homologous group (Supplementary Note 1). The non-inferiority study design was based on the immuno-bridging standards in granting Emergency Use Authorization for COVID-19 vaccine (including MVC-COV1901) by Taiwan FDA (https://www.fda.gov.tw)[20]. The immuno-bridging success criteria was the lower limit of the 2-sided 95% confidence interval (CI) for geometric mean titer (GMT) ratio >0.67[20].

The study was conducted in a single institute in Chang Gung Memorial Hospital, Linko branch, in Taiwan. After receiving the treatment, the participants remained in the study for 168 days following booster vaccination.

Written informed consent was obtained from all participants, and the trial is being done in accordance with the principles of the Declaration of Helsinki and Good Clinical Practice. This study was approved by the Taiwan Food and Drug Administration and the ethics committee at Chang Gung Medical Foundation (Taiwan). The study was registered in ClinicalTrials.gov with ID NCT05054621 and the protocol in detail is available in Supplementary information.

### Outcomes
For the study primary objective, humoral immunogenicity including serologic neutralizing antibody titers against SARS-CoV-2 and serological quantification of binding antibody to SARS-CoV-2 antigen was assessed during the duration of the study at baseline and after booster vaccination at day 10 ± 3, day 28 ± 3, day 56 ± 3 and day 168 ± 7. The SARS-CoV-2 antigen specific B cell and T cell frequencies were assessed in day 0, day 10 ± 3 and day 28 ± 3 after booster vaccination. Safety was assessed during the duration of the study. The solicited adverse events (AEs) occurring locally or systemically were assessed for 7 days following each vaccination from day 0 through day 7. Unsolicited AEs were recorded for 28 days after the boost dose. Serious AEs (SAEs) were recorded from signing of the informed consent form through day 168. Adverse events of special interest (AESIs) were recorded from the booster vaccination through day 168.

### Randomization and blinding
All eligible participants were 1:1 randomly assigned to receive a single dose of either the same vaccine as their prime dose ChAdOx1 (homologous group) or the Medigen COVID-19 vaccine MVC-COV1901 (heterologous group). Stratified randomization was used based on the intervals between prime and boost vaccination. Participants were stratified according to the prime-boost intervals of 4–6 weeks and 8–10 weeks, respectively, with equal-sized strata. Randomization was applied to each stratum and the random number list was generated by an independent study statistician using SAS software.

The treatment phase was conducted in a single-blinded fashion such that the participants were masked to the vaccine received but not to the prime-boost interval. Clinical staffs who involved in the vaccine delivery were aware of which vaccine the participant received, but the participant remained blinded by preparing the vaccine out of sight and applying a masking tape over the vaccine syringe. Laboratory staffs were also blinded to the vaccine the participant received, which may minimize the evaluation bias from the knowledge about the treatment assignment of the participant.

### Surrogate neutralizing titers by ELISA method
All serum samples were analyzed by the SARS-CoV-2 antibody ELISA kit according to the manufacturers' instructions (MeDiPro, Taiwan)[21,22].

MeDiPro was a Taiwan FDA-approved kit for quantifying Spike S1-and receptor-binding domain (RBD)-binding antibodies which were surrogates of live virus neutralization titers with high correlation[21,22]. With the cutoff of <34.47 IU/mL defining negative result, the sensitivity and specificity of the test was 92.2% (95% CI, 84.0%–96.4%) and 93% (95% CI, 81.4%–97.6%), respectively.

## Live virus neutralization assay

The neutralization assay following the standard protocol of a plaque reduction neutralization test was performed on the serum samples collected at day 28 after booster dose of vaccination. Vero cells were regularly maintained in minimal essential medium (MEM) supplemented with 10% (v/v) fetal bovine serum. Wild type virus (Wuhan strain) and Delta variant of SARS-CoV-2 were propagated in Vero cells in MEM. Serum samples were inactivated at 56 °C for 30 min before use. Serum were two-fold diluted serially and were mixed with equal volumes of SARS-CoV-2 suspension containing 100 folds of the median tissue culture infectious dose. The mixture was incubated for 2 h at 37 °C, and then an equal volume of suspended Vero E6 cells (approximately 30,000 cells/well) was added to each well. Following incubation for 1 week at 37 °C, cells were fixed with 5% glutaraldehyde and stained with 0.1% crystal violet. Serum neutralization titers were calculated and expressed as the reciprocals of the highest serum dilution that inhibits cytopathic effects.

## PBMC preparation

PBMCs were separated from heparinized blood by density gradient centrifugation using lymphoprep (Stemcell Technologies, Canada) for 20 min, 800 g at 20 °C. The PBMC band was collected, transferred to a sterile conical tube pre-filled with RPMI medium and centrifuged for 10 min, 720 g. The supernatant was discarded, the cell pellet was resuspended in RPMI medium containing 10% fetal bovine serum and centrifuged for 10 min, 400 g. Following this centrifugation, the supernatant was discarded, and the cell pellet was resuspended in RPMI medium containing 10% fetal bovine serum. An aliquot of cell suspension was used for counting and viability assessment.

## Ex vivo enzyme-linked immunospot for detection of antibody-secreting B cell response

Sterile, clear 96-well filter plate with 0.45 μm pore size hydrophobic PVDF membrane (MAIPS4510, Millipore, United States) were coated with 100 μL of 15 μg/mL Wuhan-Hu-1 strain SARS-CoV-2 spike trimer diluted in carbonate buffer or carbonate buffer only as negative control or polyvalent anti-human Ig's (Thermo Fisher Scientific, United States) as positive control and incubated overnight at 4 °C. Plates were washed with PBS three times and blocked using RPMI medium containing 10% fetal bovine serum for 1 h at 37 °C. Blocking medium was removed and plates were washed with PBS. 100 μL of PBMC suspension was added at a density of $2 \times 10^5$ cells per well for antigen-specific response and $2 \times 10^4$ cells per well for total IgG, IgM and IgA response (positive control) and incubated for 18 h at 37 °C. Cell suspension was removed and plates were washed with PBS. 100 μL of 1:5000 diluted alkaline phosphatase conjugated anti-human IgG, IgM or IgA (Calbiochem, United States) was added and incubated for 2 h at room temperature. After washing with PBS, 50 μL of BCIP/NBT-plus substrate (Mabtech, United States) was added and left for 2 to 5 minutes at room temperature. After distinct spots developed, the reaction was stopped using distilled water. Plates were air-dried and spots were measured and counted with automatic ELISpot reader.

## Ex vivo interferon-γ enzyme-linked immunospot for detection of cellular response

Interferon-γ ELISpot assay was performed using the human IFN-γ ELI-Spot basic kit (ALP) (3420-2 A, Mabtech, United States) according to the manufacturer's protocol with some modifications. Sterile, clear 96-well filter plate with 0.45 μm pore size hydrophobic PVDF membrane (MAIPS4510, Millipore, United States) was coated by 100 μL of 15 μg/mL anti-human IFN-γ monoclonal antibody (1-D1K) (Mabtech, United States) in sterile PBS overnight at 4 °C. Plates were then washed with PBS five times and blocked with 250 μL per well of RPMI medium containing 10% fetal bovine serum for 1 h at 37 °C. After removing the medium, 100 μL of PBMC suspension was added to each well, and SARS-CoV-2 spike S1 subunit or S2 subunit peptide pool (PP003, Sino Biological, China) at a final concentration of 2 μg/ml was added. Control wells of cells incubated with 0.4% DMSO (negative control), or with 10 μg/mL phytohemagglutinin (PHA) (positive control) were also included. After incubation for 18 h at 37 °C, plates were washed with PBS six times and incubated with 100 μL of 1 μg/mL biotinylated anti-human IFN-γ antibody (7-B6-1) (Mabtech, United States) in sterile PBS with 5% BSA for 2 h at room temperature. Plates were then washed with PBS six times and incubated with 50 μL of 1:1000 dilution of Streptavidin-Alkaline phosphatase in sterile PBS for 1 h at room temperature. Plates were then washed with PBS three times and developed with 100 μL of substrate solution (Mabtech, United States) until distinct spots emerged. The color development was terminated by washing plates with distilled water. Plates were air-dried and spots were measured and counted with an automatic ELISpot reader.

## Sample size

The primary objective of this trial was to determine if the immune response of heterologous group was non-inferior to that observed in homologous group, and the primary endpoints was neutralizing antibody titer at day 28 after booster vaccination. By assuming the non-inferiority margin was 0.67-fold-difference or −0.401 absolute difference of log GMT between heterologous group and homologous group with the standard deviation 0.66, and the true difference of log GMT was 0, the study needed to recruit 44 evaluable participants per group (total 88 participants) to achieve 80% of power at one-sided 2.5% significance level. According to the missing rate 10% and the stratification in 1:1 for prime-boost 4–6 and 8–10 weeks, 50 participants for each stratum and equally random assigned to each group (25 for heterologous and 25 for homologous group) within strata (total 100 participants) is needed. The mean difference of log GMT was presented with the two-sided 95% CI.

## Statistical analysis

Descriptive statistics on continuous measurements included means, medians, standard deviations, and ranges, while categorical data was summarized using frequency counts and percentages. For the immunogenicity endpoints including SARS-CoV-2 neutralizing antibody levels and cell-mediated immune responses, the point estimates were reported with 95% confidence intervals. For the secondary endpoints for comparisons of continuous scale between groups, independent t-test was used. For the comparison of nAb titer changes from baseline between groups of short and long prime-boost intervals, a value of 1 was used to substitute a change value with zero or a value less than zero, so that the information contained in these data was not lost in calculation of geometric mean.

For safety analysis, the number (%) of subjects with AEs was reported. Frequency counts and percentages were also be presented of subjects with serious AEs, AEs leading to withdrawal, AEs by severity and AEs by relationship to study treatment. All other safety measures were analyzed for the safety population.

Evidence of significant interaction was assessed at the 5% level. All analyses were performed using the Statistical Analysis System (SAS) statistical software package, version 9.4. (SAS Institute Inc, Cary, NC) and Graphpad Prism (Version 9.1.1, GraphPad Software, US).

### Interim report and enrolment status

The study duration for each participant would be nearly or more than 6 months following the enrolment (visit day $-70\sim-1$, 0, 7, $10 \pm 3$, $28 \pm 3$, $56 \pm 3$ and $168 \pm 7$, Supplementary Note 1). This analysis was prospectively specified in the protocol (Supplementary Note 1) and reported based on the data collected from all enrolled participants until day 28. As of 01 February 2022, a total of 101 subjects were enrolled in the study. One subject failed screening because he was unable to visit the study site in the scheduled time points. The remaining 100 were followed up, and 0 completed the study when this analysis was reported. The earliest vaccination campaign start date was 15 September 2021.

### Reporting summary

Further information on research design is available in the Nature Research Reporting Summary linked to this article.

## Data availability

The data associated with this study are available within the article, its supplementary information and Source Data file. This trial is registered on ClinicalTrials.gov under the identifier NCT05054621. Source data are provided with this paper.

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

## Acknowledgements

The authors acknowledge the assistance provided by the Clinical Trial Center, Chang Gung Memorial Hospital, Taoyuan, Taiwan, which was funded by the Ministry of Health and Welfare of Taiwan (MOHW110-TDU-B-212-124005). The trial was partly supported by funding (CPRPG3L0041, CJC) from Medigen Vaccine Biologics Corp in Taiwan. This work was partly supported by the Chang Gung Memorial Hospital (BMRPE22) and the Ministry of Science and Technology of Taiwan (MOST 110-2628-B-182-013) to K.-Y.A.H. The funder has no role in the design and conduct of the study, nor the decision to prepare and submit the manuscript for publication.

## Author contributions

K.-Y.A.H. conceived the study. K.-Y.A.H. and C.-J.C designed the study and produced the protocol. L.-Y.Y. and W.-Y.C. assisted statistical design and analysis of the study. K.-Y.A.H. designed and performed B and T cell experiments. C.-G.H. designed and performed serological experiments. Y.-C.H., C-H.-C., and S.-R.S. helped prepare materials, perform experiments and analyse data. K.-Y.A.H. and C.-J.C. wrote the first draft of manuscript. All authors read and approved the manuscript.

## Competing interests

The authors declare no competing interests.

## Additional information

[1]Division of Pediatric Infectious Diseases, Departments of Pediatrics, Chang Gung Memorial Hospital, 333 Taoyuan, Taiwan, ROC. [2]School of Medicine, College of Medicine, Chang Gung University, 333 Taoyuan, Taiwan, ROC. [3]Molecular Infectious Diseases Research Center, Chang Gung Memorial Hospital, 333 Taoyuan, Taiwan, ROC. [4]Clinical Trial Center, Chang Gung Memorial Foundation, 333 Taoyuan, Taiwan, ROC. [5]Research Center for Emerging Viral Infections, College of Medicine, Chang Gung University, 333 Taoyuan, Taiwan, ROC. [6]Department of Laboratory Medicine, Linkou Chang Gung Memorial Hospital, 333 Taoyuan, Taiwan, ROC. [7]Department of Medical Biotechnology and Laboratory Science, College of Medicine, Chang Gung University, 333 Taoyuan, Taiwan, ROC. [8]Research Center for Chinese Herbal Medicine, Research Center for Food and Cosmetic Safety, and Graduate Institute of Health Industry Technology, College of Human Ecology, Chang Gung University of Science and Technology, 333 Taoyuan, Taiwan, ROC. [9]Genomics Research Center, Academia Sinica, 115 Taipei, Taiwan, ROC. ✉e-mail: joyce@cgmh.org.tw; arthur1726@cgmh.org.tw

