## [Peer review file · Nature Communications]

REVIEWER COMMENTS

Reviewer #1 (Remarks to the Author):

-First let me congratulate the team on having written an outstanding scientific account of the study with perfect English. It was a pleasure to read

-The MVC-CoV1901 vaccine has demonstrated excellent results and is being deployed for use as the first of its class (protein S2P based vaccine) authorized in the world. I think this should be stated on the paper

-In Methods I would clarify that enrolled participants did not have history to TTS and had not experienced any severe adverse events after first dose of AZ vaccine

-Page 5, please also clarify here the interval between doses

-Page 14 I would strongly consider evaluating CD8 responses in particular, as you might find a valuable piece of data to compare vaccines

-Fig 2, would it be possible to compare the reactogenicity to published work with 2 doses of MVC-CoV1901?

-Fig 3, include HCS or mention average IU for comparison

-Fig 3 and discussion, when you assay for RBD and S1, could you comment on possible contribution of S2P compared to wild type spike on immunogenicity of each vaccine?

-Discussion, I would compare results with widely available data of first AZ dose followed by Pfizer, Moderna, and inactivated vaccines, especially PsvN titers and reactogenicity

-are you planning on other designs of mix-match between this vaccine and others? especially mRNA?

-do you have plans for any variant or multivalent vaccine studies?

Reviewer #2 (Remarks to the Author):

Review of "Immunogenicity and reactogenicity of heterologous ChAdOx1 nCoV-19 and an adjuvanted recombinant S-2P subunit vaccine MVC-COV1901 against COVID-19: interim analysis of a randomized control trial" (NCOMMS 22-06522) by Chen et al

General Comments: While the primary results appear very clear-cut, showing an advantage of the heterologous arm, the paper is not as clear as it could be about these results. While the abstract is technically correct that the non-inferiority test was met, the abstract should also indicate that the heterologous arm was shown to be superior, and this should be clear in the results section as well.

1. It is unfortunate that demonstration of potential superiority did not appear to be planned for in the protocol; nonetheless, it is better to communicate this finding to the reader early than to

wait until the Discussion section which appears to be how this was handled. Given there was some expectation that the heterologous arm would be better before the study began, there

should have been a clear plan to test for superiority and if that is not met, then test for non inferiority. Thankfully, there is no meaningful statistical concern associated with doing this

maneuver post hoc. So, it is better to inform the reader of this finding, despite the lack of pre specified details associated with it.

2. It is puzzling why the title refers to an interim analysis. There is no indication that this analysis is from an interim analysis, as the sample size calculation is for 50 subjects per arm, which appears to have been completed. Perhaps the authors were trying to convey that this is a early stage study, but if so, interim analysis is not the proper term.

3. While the non-inferiority margin is not consequential to the result, there should have been some brief rationale provided for its magnitude.

4. The protocol states that “The primary analysis will be conducted on the modified intent-to-treat basis among the participants received a boost dose, i.e. we will only include participants whose primary endpoint at D28 post boost is available.” The phrase about availability does not appear

to match with the definition of the modified ITT approach. If there were missing data, and the analysis is conducted only on complete cases, that should be clear. Furthermore, the paper

notes that (unspecified) methods will be used for missingness. The Consort diagram suggests there was no missing data, but it is not completely clear that this is the case. If there was no

missing data for the primary endpoint, that should be simply stated. If there were missing data, then the methods for handling it should be spelled out somewhere.

5. Why was the study only single-blind? What exactly does this mean? What were the consequences of this?

6. The sample size section states that “adjusted mean difference of log GMT was presented”. However, nowhere else refers to adjusted differences, nor was the nature of the adjustment explained. This needs to be clarified.

7. The authors observed that the subjects with short term boost had higher responses than those with long term boost, but they did not seem to take into account the differences at baseline in

these strata. A more meaningful endpoint or analysis would be based on difference from baseline.

8. The paper states that “Participants in each study arm were equally divided into two subgroups according to the intervals of 4-6 weeks and 8-10 weeks, respectively, between the prime and boost doses.” As a minor comment: while it is presumed that this different timing was just a matter of when an individual enrolled in the study relative to their last vaccine, as opposed to some sort of randomized intervention, the use of “equally divided” almost sounds like it was something the study did. So, it is recommended that this be described more clearly.

Dear Editor and Referees,

This letter is in response to the valuable suggestions for the revision of the paper “Immunogenicity and reactogenicity of heterologous ChAdOx1 nCoV-19 and an adjuvanted recombinant S-2P subunit vaccine MVC-COV1901 against COVID-19: interim analysis of a randomized control trial”. We have made revisions in response to these helpful comments. We think all the comments are very important to us, and we have taken each point into consideration. Please refer to the following responses.

Reviewer #1 (Remarks to the Author):

-First let me congratulate the team on having written an outstanding scientific account of the study with perfect English. It was a pleasure to read

-The MVC-CoV1901 vaccine has demonstrated excellent results and is being deployed for use as the first of its class (protein S2P based vaccine) authorized in the world. I think this should be stated on the paper

A line has been added in the introduction to state that MVC-CoV1901 was the first S-2P protein-based vaccine being deployed against COVID-19 in the world (lines 12, 13, page 5).

-In Methods I would clarify that enrolled participants did not have history to TTS and had not experienced any severe adverse events after first dose of AZ vaccine

Yes, they were all free of TTS or other serious AE following the first dose of ChAdOx1 vaccination. A line has been added in lines 24-25, page 5. 'There was no TTS or other serious AE following the first ChAdOx1 vaccination in all participants.'

-Page 5, please also clarify here the interval between doses

We have revised the section of 'Randomization and blinding' and further clarified the interval between doses in the revised manuscript (page 7). Stratified randomization was used based on the intervals between prime and boost vaccination. Participants were stratified according to the prime-boost intervals of 4-6 weeks and 8-10 weeks, respectively, with equal-sized strata. Randomization was applied to each stratum and the random number list was generated by an independent study statistician using SAS software.

-Page 14 I would strongly consider evaluating CD8 responses in particular, as you might find a valuable piece of data to compare vaccines

We agree that the contribution of CD8 T cells to overall spike-specific T cell response is unclear and intriguing in the study; nevertheless, we did not store cryopreserved PBMCs at that time and no samples are available for further cellular assays. Here, freshly separated PBMCs were used in the *Ex vivo* enzyme-linked immunospot assays in the study. We have further discussed T cell response in the revised manuscript (page 20).

-Fig 2, would it be possible to compare the reactogenicity to published work with 2 doses of MVC-CoV1901?

Yes, we have briefly mentioned the comparisons of safety data in this trial and other published papers in the Discussion section (references 4, 5, 16, 23; page 17). The reactogenicity profiles in this trial were generally consistent with the safety data published for the homologous schedule of both vaccines in their respective clinical trials. However, a direct comparison of different trials may not be appropriate given their different study designs and different age groups of enrolled participants.

-Fig 3, include HCS or mention average IU for comparison

Figure 3A has been revised and the data from 15 human convalescent serum (HCS) samples collected at day 28±3 days of diagnosis has been added to this figure. The legend of the figure is also revised (page 28).

-Fig 3 and discussion, when you assay for RBD and S1, could you comment on possible contribution of S2P compared to wild type spike on immunogenicity of each vaccine?

The advantage of S-2P is briefly described in the Introduction section of revised manuscript (page 5). Multiple lines of evidence have demonstrated the superiority of stabilized prefusion conformation of spike protein by the addition of 2P and furine cleavage site mutation in the immunogenicity and protective efficacy in animal models (page 5; references 17, 18).

-Discussion, I would compare results with widely available data of first AZ dose followed by Pfizer, Moderna, and inactivated vaccines, especially PsvN titers and reactogenicity

Thank you for the suggestion. A paragraph has been added to discuss the heterologous schedule involving different COVID-19 vaccines (page 18).

-are you planning on other designs of mix-match between this vaccine and others? especially mRNA?

Yes, a study evaluating the immunogenicity and reactogenicity of heterologous regimen with MVC-COV1901 and vaccines of other platforms including mRNA and adenovirus vector is ongoing in our institute.

-do you have plans for any variant or multivalent vaccine studies?

A protein-based vaccine using the S-2P of Beta variant is under evaluation in phase I clinical trial In Taiwan by Medigen. Our group does not plan clinical studies for other variant or multivalent vaccines at this moment.

Reviewer #2 (Remarks to the Author):

Review of “Immunogenicity and reactogenicity of heterologous ChAdOx1 nCoV-19 and an adjuvanted recombinant S-2P subunit vaccine MVC-COV1901 against COVID-19: interim analysis of a randomized control trial” (NCOMMS 22-06522) by Chen et al

General Comments: While the primary results appear very clear-cut, showing an advantage of the heterologous arm, the paper is not as clear as it could be about these results. While the abstract is technically correct that the non-inferiority test was met, the abstract should also indicate that the heterologous arm was shown to be superior, and this should be clear in the results section as well.

We have further revised the abstract to highlight stronger spike-specific immune responses elicited in the heterologous group than those in the homologous group (page 3). The nAb GMT ratio between heterologous and homologous groups and the fold increase in the nAb titer for the heterologous group are clearly indicated in the abstract of revised manuscript (page 3). The related description has also been revised in the Results section (pages 13, 15).

1. It is unfortunate that demonstration of potential superiority did not appear to be planned for in the protocol; nonetheless, it is better to communicate this finding to the reader early than to wait until the Discussion section which appears to be how this was handled. Given there was some expectation that the heterologous arm would be better before the study began, there should have been a clear plan to test for superiority and if that is not met, then test for noninferiority. Thankfully, there is no meaningful statistical concern associated with doing this maneuver post hoc. So, it is better to inform the reader of this finding, despite the lack of prespecified details associated with it.

We have further addressed significantly stronger spike-specific immune responses elicited in the heterologous group than those in the homologous group in the Abstract, Results and first paragraph of Discussion sections in the revised manuscript (pages 3, 14, 15, 18).

When we conceived and designed the study last year, limited data were available on the immunogenicity of MVC-COV1901 vaccine and the antibody response to heterologous SARS-CoV-2 vaccine regimens. Therefore, the study was designed as a

non-inferiority trial and the Taiwan FDA regulatory requirement was utilized to determine the non-inferiority margin and sample size at that time. We have also mentioned that it lacks a preplanned definition of superiority that allowed for a switch from non-inferiority to superiority in the trial protocol (page 6), while clinical trial should be based on a protocol that details the study rationale, proposed methods, organization, and ethical considerations.

2. It is puzzling why the title refers to an interim analysis. There is no indication that this analysis is from an interim analysis, as the sample size calculation is for 50 subjects per arm, which appears to have been completed. Perhaps the authors were trying to convey that this is an early stage study, but if so, interim analysis is not the proper term.

We have further clarified this issue in the section of 'Interim analysis and enrollment status' in the revised manuscript (page 11). The study duration for each participant would be nearly or more than 6 months following the enrolment (visit day -70~-1, 0, 7, 10±3, 28±3, 56±3 and 168±7, Supplementary Data 1). This interim analysis was prospectively specified in the protocol and based on the data collected from all enrolled participants until day 28. All participants were still followed up when this interim analysis was reported.

3. While the non-inferiority margin is not consequential to the result, there should have been some brief rationale provided for its magnitude.

We have further clarified this issue in the section of 'Study design' in the revised manuscript (page 6). When we conceived and designed the study last June, limited data were available on the immunogenicity of MVC-COV1901 vaccine and the antibody response to heterologous SARS-CoV-2 vaccine regimens. Therefore, the Taiwan FDA regulatory requirement was utilized to determine the non-inferiority margin and sample size at that time.

The non-inferiority study design was based on the immuno-bridging standards in granting Emergency Use Authorization for COVID-19 vaccine (including MVC-COV1901) by Taiwan FDA (<https://www.fda.gov.tw>). The immuno-bridging success criteria was the lower limit of the 2-sided 95% confidence interval for geometric mean titer ratio >0.67 (reference 20).

4. The protocol states that “The primary analysis will be conducted on the modified intent-to-treat basis among the participants received a boost dose, i.e. we will only include participants whose primary endpoint at D28 post boost is available.” The phrase about availability does not appear to match with the definition of the modified ITT approach. If there were missing data, and the analysis is conducted only on complete cases, that should be clear. Furthermore, the paper notes that (unspecified) methods will be used for missingness. The Consort diagram suggests there was no missing data, but it is not completely clear that this is the case. If there was no missing data for the primary endpoint, that should be simply stated. If there were missing data, then the methods for handling it should be spelled out somewhere.

In the vaccine trial, the study duration for each participant would be nearly or more than 6 months following the enrolment (visit day -70~-1, 0, 7, 10±3, 28±3, 56±3 and 168±7, Supplementary Data 1). All of the 100 enrolled participants completed the D28 visit without missing data when this interim analysis was reported. We have further clarified this issue in the section of ‘Interim analysis and enrollment status’ in the revised manuscript (page 11).

5. Why was the study only single-blind? What exactly does this mean? What were the consequences of this?

We have further clarified this issue in the section of “Randomization and blinding” in the revised manuscript (page 7).

The treatment phase was conducted in a single-blinded fashion such that the participants were masked to the vaccine received but not to the prime-boost interval. Clinical staffs who involved in the vaccine delivery were aware of which vaccine the participant received, but the participant remained blinded by preparing the vaccine out of sight and applying a masking tape over the vaccine syringe. Laboratory staffs were also blinded to the vaccine that the participant received, which may minimize the evaluation bias from the knowledge about the treatment assignment of the participant.

6. The sample size section states that “adjusted mean difference of log GMT was presented”. However, nowhere else refers to adjusted differences, nor was the nature of the adjustment explained. This needs to be clarified.

We have removed “adjusted” in the related description to avoid the confusion (page 11).

In the trial, the primary endpoint was neutralizing titer measured at day 28 post boosting. The GMT was compared between heterologous and homologous boost groups under the hypothesis:

H0: GMT heterologous / GMT homologous ≤ 0.67 or \log GMT heterologous - \log GMT homologous ≤ -0.401 ;

H1: GMT heterologous / GMT homologous > 0.67 or \log GMT heterologous - \log GMT homologous > -0.401 .

The GMT was transferred using logarithmic transformations to render a normal distribution. The mean difference of \log GMT was presented with the two-sided 95% confidence interval. We claimed that heterologous boost group was non-inferior to homologous boost group if the lower confidence interval lies above -0.401.

7. The authors observed that the subjects with short term boost had higher responses than those with long term boost, but they did not seem to take into account the differences at baseline in these strata. A more meaningful endpoint or analysis would be based on difference from baseline.

We agree that there was a trend toward a relatively higher antibody titer at baseline in the subgroup of 4-6 weeks interval compared with that of 8-10 weeks interval. This observation is in accord with previous findings that anti-spike antibody titer would peak at approximately 4-6 weeks and gradually decline after one dose of ChAdOx1 (reference 26; Flaxman et al., 2021 doi: 10.1016/S0140-6736(21)01699-8). Although such trend of baseline antibody titer was observed in the study, in the heterologous MVC-COV1901 group, the difference of baseline antibody titer between two prime-boost interval strata did not reach statistical significance. After boosting with MVC-COV1901, the recipients with short prime-boost interval (4–6 weeks) had significantly higher antibody titers compared to those with long interval (8–10 weeks) at day 10 \pm 3 and day 28 \pm 3. In the homologous group, the baseline antibody titer in the subgroup of 4-6 weeks interval is higher than that in the subgroup of 8-10 weeks interval. After boosting with ChAdOx1, a similar finding of higher antibody titers favoring the short interval was also identified in the ChAdOx1 group at day 10 \pm 3. A consistent observation on the enhanced immunogenicity for the heterologous schedule was noted when the prime and boost vaccines were administered at a short interval between 4 to 6 weeks, although the underlying mechanism remains unclear (page 17).

We also compared baseline antibody titer between heterologous and homologous groups for each prime-boost interval stratum and there was no significant difference in the baseline antibody titer between two groups. The statistical analysis is provided as follows,

prime-boost interval strata	MVC-COV1901		ChAdOx1		MVC-COV1901 /ChAdOx1	p value
	n	GMT mean , 95%CI	n	GMT mean , 95%CI	GMT ratio , 95%CI	
4-6 weeks, Day 0	25	39.0 (23.5, 64.8)	25	43.5 (25.1, 75.4)	0.9 (0.4, 1.9)	0.764
8-10 weeks, Day 0	25	26.6 (16.2, 43.7)	25	21.0 (14.5, 30.3)	1.3 (0.7,2.3)	0.431

We have further provided related information in the revised manuscript (Page 13, Supplementary Table 6).

- The paper states that “Participants in each study arm were equally divided into two subgroups according to the intervals of 4-6 weeks and 8-10 weeks, respectively, between the prime and boost doses.” As a minor comment: while it is presumed that this different timing was just a matter of when an individual enrolled in the study relative to their last vaccine, as opposed to some sort of randomized intervention, the use of “equally divided” almost sounds like it was something the study did. So, it is recommended that this be described more clearly.

We have further clarified the randomization method in the revised manuscript (page 7). Stratified randomization was used based on the intervals between prime and boost vaccination. Participants were stratified according to the prime-boost intervals of 4-6 weeks and 8-10 weeks, respectively, with equal-sized strata. Randomization was applied to each stratum and the random number list was generated by an independent study statistician using SAS software.

REVIEWERS' COMMENTS

Reviewer #2 (Remarks to the Author):

Review of NCOMMS-22-06522A

By Chen et al.

While the revision has improved the manuscript, some outstanding issues remain that should be straightforward to address:

General Comments & Item 1: As noted in the US FDA Guidance on Non-inferiority trials there is no statistical problem noting superiority once the planned non-inferiority is met (the real problem occurs under the completely reverse scenario where superiority is not met and a non-inferiority margin had not been pre-specified). In any event, it should be conveyed to the reader that the result is stronger than non-inferiority, which sounds like the regimens are “similar”. The authors appear to be concerned that superiority had not been defined in the protocol. Thus, it is fine that they still use the term non-inferiority as this reflects the design, and if they prefer, they can avoid the term “superiority”, but it is very much recommended that the significantly higher result on the primary endpoint be noted right when presenting the primary endpoint confidence intervals. While the end of the abstract now says “significantly higher” in a concluding statement, this could be confusing after the weaker “non-inferiority” statements.

(While not noted in the previous review: if not currently mentioned in the paper, there should be some comment somewhere about why two primary analyses were presented (wild-type and Delta) since not in the protocol. There is no problem with presenting this dual result, and it is understood that some things cannot be pre-specified in a highly dynamic setting such as Covid, but it should just be clarified. Similarly, sometimes the paper just reports results on neutralizing antibodies, but does not refer to the variant; this should be clearer given the two variants reported for the primary analysis.)

Item 2: It is now understood what the authors meant by “interim analysis”, however this is not the common use of this term, which implies that the study was stopped early because of the nature of the partial results of the primary endpoint. It is recommended that any reference to “interim analysis” be omitted, with just a mention in the design section that the study was pre-planned to report the primary endpoint results while the study was ongoing to complete long term follow-up. Alternatively, if the authors want to use a phrase, they could use “Interim report”, but the previous solution seems more standard.

Item 3: The authors still have not compared change from baseline when comparing short interval and long interval, and it is not sufficient to just test for significant difference at baseline. Thus, it is still recommended that the authors have a figure (maybe Figure 4b) that is like the current Figure 4 but looks at relative change from baseline to Day 28 (or whatever is the appropriate metric). Furthermore,

to better justify the statement in the abstract about greater benefit of heterologous regimen in the short interval stratum, it would be helpful to provide an estimate of the treatment effect within each stratum.

Dear Editor and Referees,

This letter is in response to the valuable suggestions for the revision of the paper. The title was modified to fulfill the recommendations of the CONSORT statement and the format of *Nature Communications*. The revised title is “**A randomized controlled trial of heterologous ChAdOx1 nCoV-19 and recombinant subunit vaccine MVC-COV1901 against COVID-19**”. We have also revised main text in response to helpful comments. We think all the comments that are very important to us, and we have taken all these points into consideration. Please refer to the following responses.

General Comments & Item 1: As noted in the US FDA Guidance on Non-inferiority trials there is no statistical problem noting superiority once the planned non-inferiority is met (the real problem occurs under the completely reverse scenario where superiority is not met and a non-inferiority margin had not been pre-specified). In any event, it should be conveyed to the reader that the result is stronger than non-inferiority, which sounds like the regimens are “similar”. The authors appear to be concerned that superiority had not been defined in the protocol. Thus, it is fine that they still use the term non-inferiority as this reflects the design, and if they prefer, they can avoid the term “superiority”, but it is very much recommended that the significantly higher result on the primary endpoint be noted right when presenting the primary endpoint confidence intervals. While the end of the abstract now says “significantly higher” in a concluding statement, this could be confusing after the weaker “non-inferiority” statements.

(While not noted in the previous review: if not currently mentioned in the paper, there should be some comment somewhere about why two primary analyses were presented (wild-type and Delta) since not in the protocol. There is no problem with presenting this dual result, and it is understood that some things cannot be pre-specified in a highly dynamic setting such as Covid, but it should just be clarified. Similarly, sometimes the paper just reports results on neutralizing antibodies, but does not refer to the variant; this should be clearer given the two variants reported for the primary analysis.)

Response: Thanks for the comment. We have realized that it is acceptable to declare superiority in a non-inferiority trial if the lower limit of CI of the study treatment is above the non-inferiority margin and above zero, because it usually takes into account and controls for the Type I error and does not penalize for multiple testing. We have revised the Abstract and pointed out that at day 28 post-boosting, the neutralizing antibody geometric mean titer against wild-type SARS-CoV-2 in MVC-COV1901 recipients (236 IU/mL) was superior to that in ChAdOx1 recipients (115 IU/mL), with a GMT ratio of 2.1 (95% CI, 1.4 to 2.9) and superiority in the neutralizing antibody titer against Delta variant was also found for heterologous MVC-COV1901 immunization with a GMT ratio of 2.6 (95% CI, 1.8 to 3.8) (lines 8–13, page 3)

The SARS-CoV-2 Delta variant was the dominant strain circulating globally in the second half of year 2021 when the clinical trial was conducted. Thus, we further look into in the immunogenicity of vaccines to this circulating Delta variant in addition to the ancestral Wuhan strain. We have further clarified this issue in the Introduction of revised manuscript (lines 5-6, page 5).

Item 2: It is now understood what the authors meant by “interim analysis”, however this is not the common use of this term, which implies that the study was stopped early because of the nature of the partial results of the primary endpoint. It is recommended that any reference to “interim analysis” be omitted, with just a mention in the design section that the study was pre-planned to report the primary endpoint results while the study was ongoing to complete long term follow-up. Alternatively, if the authors want to use a phrase, they could use “Interim report”, but the previous solution seems more standard.

Response: Thanks for the comment. We have removed the term ‘interim analysis’ from the title and the main text. The ‘Interim report’ is used as subtitle in the Methods (page 21).

Item 3: The authors still have not compared change from baseline when comparing short interval and long interval, and it is not sufficient to just test for significant difference at baseline. Thus, it is still recommended that the authors have a figure (maybe Figure 4b) that is like the current Figure 4 but looks at relative change from baseline to Day 28 (or whatever is the appropriate metric). Furthermore, to better justify the statement in the abstract about greater benefit of heterologous regimen in the short interval stratum, it would be helpful to provide an estimate of the treatment effect within each stratum.

Response: We have further analyzed nAb titer changes from baseline between groups of short and long vaccination regimens and provided the data in the revised manuscript (pages 8 and 20; Supplementary Figure 1). We have modified the abstract in the revised manuscript (page 3).